artificial intelligence/pattern recognition

COVID-19 demographics impacts, COVID-19 symptoms, VOC 202012/01, rule mining in COVID-19, global deaths in COVID-19, patterns analysis in COVID-19 data

# Analysing the impact of global demographic characteristics over the COVID-19 spread using class rule mining and pattern matching

Wasiq Khan[1], Abir Hussain[1], Sohail Ahmed Khan[2],
Mohammed Al-Jumailey[3], Raheel Nawaz[4]
and Panos Liatsis[5]

[1]Department of Computing and Mathematics, Liverpool John Moores University, Liverpool L33AF, UK
[2]Department of Computer Science, DeepCamera Research Lab, Interactive Media, Smart System, and Emerging Technologies Center, Nicosia, Cyprus
[3]The Regenerative Clinic, Queen Anne Medical Centre, Harley Street Medical Area, London
[4]Department of Computing and Mathematics, Manchester Metropolitan University, Manchester M156BH, UK
[5]Department of Electrical Engineering and Computer Science, Khalifa University, PO Box 127788, Abu Dhabi, UAE

WK, 0000-0002-7511-3873; RN, 0000-0001-9588-0052

**Author for correspondence:**
Wasiq Khan
e-mail: w.khan@ljmu.ac.uk

Since the coronavirus disease (COVID-19) outbreak in December 2019, studies have been addressing diverse aspects in relation to COVID-19 and Variant of Concern 202012/01 (VOC 202012/01) such as potential symptoms and predictive tools. However, limited work has been performed towards the modelling of complex associations between the combined demographic attributes and varying nature of the COVID-19 infections across the globe. This study presents an intelligent approach to investigate the multi-dimensional associations between demographic attributes and COVID-19 global variations. We gather multiple demographic attributes and COVID-19 infection data (by 8 January 2021) from reliable sources, which are then processed by intelligent algorithms to identify the significant associations and patterns within the data. Statistical results and experts' reports indicate strong associations between COVID-19 severity levels across the globe and certain demographic attributes, e.g. female smokers, when combined together with other attributes. The outcomes will aid the understanding of the

dynamics of disease spread and its progression, which in turn may support policy makers, medical specialists and society, in better understanding and effective management of the disease.

# 1. Introduction

Respiratory viral illnesses are allied with the continuing and serious psychopathological concerns among survivors [1]. Coronaviruses are ribonucleic acid (RNA) viruses that can trigger contamination illnesses, including common colds or even serious concerns such as severe acute respiratory conditions [2]. Research studies indicated that the exposure to coronavirus has shown to be associated with neuropsychiatric diseases, including Middle East respiratory syndrome (MERS), severe acute respiratory syndrome (SARS) and other outbreaks [3]. Coronavirus disease (COVID-19), which initially appeared in Wuhan, China in December 2019, is triggered by acute respiratory syndrome and is referred to as coronavirus-2 (SARS-CoV-2). In March 2020, the classification of COVID-19 was altered from a 'public health emergency' to a pandemic by WHO. The COVID-19 pandemic is the most important global health disaster in modern history and the greatest trial humans confronted since World War II, spanning every continent apart from Antarctica. There are more than 90 million cases and more than 1.9 million deaths to date (8 January 2021). Studies reported COVID-19 affects people who have a weak immune system, such as the elderly and vulnerable people with underlying medical conditions, including diabetes and cardiovascular disease (CVD). On the other hand, the effects of the virus on children and young adults are not yet fully understood, since the number of infections or/ and death rate is relatively low [4].

Various research studies addressed medical symptoms, personal attributes and demographic characteristics, which are highly correlated with the COVID-19 infection. For instance, the Centers for Disease Control and Prevention (CDC) indicated that there were 52 166 deaths in 47 US jurisdictions between 12 February to 18 May 2020 [5]. Among the decedents, the majority were found to be aged greater than or equal to 65 years, with higher ratio of males, white ethnicity while comparatively lower ratio of black, Hispanic/Latino and Asian ethnic background. Median decedent age was found to be 78 years. Authors reported that a higher percentage of Hispanic and non-white decedents were aged less than 65, compared with lower percentage of white, non-Hispanic decedents. Studies also indicated other clinical attributes, specifically, obesity [6,7], CVD and hypertension [6,8] as important factors affecting the COVID-19 infection rate. On the other hand, studies address demographic attributes such as GDP ratio of a country, smoking prevalence and average annual temperature of a country [5,6,9,10], etc. being highly correlated with the COVID-19 infection around the world.

While the aforementioned studies have identified some clinical and economic demographic parameters to predict disease spread and its associations, most of the works are either carried out at early stages with insufficient amount of data, or using conventional statistical approaches, which are limited to investigate only the individual attributes' associations with COVID-19 infection. An intelligent algorithm is needed to model the complex and multi-dimensional attributes and investigate the combined impact of various demographic characteristics over the COVID-19 severity, particularly, at the current stage, where sufficient data is available. This would support understanding of the in-depth demographic aspects of this disease, and significantly contribute towards effective policy-making and disease management.

In order to explore COVID-19 severity and its associations to multiple demographical characteristics across the globe, this study investigates *whether the diversity in COVID-19 infection severity (e.g. variations in death rate) across the globe is significantly associated with an individual or combination of demographic attribute/s?*

To answer the underlying research question, the authors have undertaken this study to model the associations between multiple demographic attributes, including economic, socio-economic, environmental and health related. The varying nature of COVID-19 infections in the global geographical context is far from clear and, therefore, adopting an open-minded approach is useful in unravelling such a complex problem. Deploying machine intelligence approaches offers an advantage over conventional statistical methods in analysing the complex patterns and potential associations between multiple predefined demographic facts and COVID-19 spread in the world. The major contributions of this study include:

— Using class association rules (CARs) to investigate the combined demographic attributes that are significantly associated with COVID-19 infection severity across the globe.

— Using self-organizing maps (SOM) for pattern identification within the multi-dimensional demographic and COVID-19-related datasets as well as detailed country-level information in the form of two-dimensional visualizations of COVID-19 spread across the globe, which is easily understandable and interpretable by humans.

— Gathering a larger COVID-19 dataset (over a period of 1 year) and various demographic characteristics from reliable public data sources and transforming them into an appropriate form using statistical approaches and medical experts' recommendations, where appropriate.

The remainder of this paper is organized as follows. Section 2 describes the existing works related to COVID-19 spread and correlated attributes. Section 3 presents the details of the proposed methodology. Experimental results and interpretation of representative rules are reported in §4, followed by the discussion of the findings, and finally, the conclusion and future directions are presented in §5.

## 2. Related work

Since the COVID-19 outbreak, research studies have been attempting to address diverse aspects of the disease, specifically, the predictive symptoms and associated attributes. Various clinical and demographic attributes were identified as potentially associated with the COVID-19 spread in different parts of the world. Research carried out in [6,8] indicated that certain male patients aged between 40 and 60 having underlying medical conditions, such as hypertension, CVD and chronic lung disease, were in a critical condition on admission, and progressed rapidly to death within two to three weeks from contracting COVID-19. Likewise, [9] reported that male patients aged over 65 years, who smoke, might face a higher risk of developing critical conditions of COVID-19. Obesity and smoking were also associated with increased risk of COVID-19 infection [6]. Study [7] also indicated obesity as an important risk factor for COVID-19 hospital admissions in patients younger than 60 years.

On the other hand, research outcomes from these studies contradict each other, specifically, in terms of demographic aspects. For example, the authors in [10] suggested that countries with a higher smoking rate had lower frequency of critical cases and deaths, whereas [6,9] indicated that high smoking rate is associated with increased risk of COVID-19 infection. The outcomes from [6] also reported other indicators, such as gender, being influential on the disease spread. Likewise, patients with high lactate dehydrogenase levels require thorough observation and early mediation to avoid the possibility of developing severe COVID-19 [11]. Male patients with heart injury, hyperglycaemia and high-dose corticosteroid use may have a high risk of death [11].

The authors in [12] suggested that children of all ages seemed susceptible to COVID-19, irrespective of gender. While COVID-19 cases in children were less severe than those of adult patients, young children, specifically infants, were found to be easily infected [12]. On the other hand, findings in [13] suggested that children may be less vulnerable to COVID-19 because children have: (i) a more active immune response, (ii) stronger respiratory tracts, since they are less exposed to cigarette smoke and air pollution in comparison with adults, and (iii) fewer underlying medical disorders. A similar study reported milder disease progression and better prognosis in children as compared with adults, with deaths being extremely rare in children [14]. On the other hand, WHO [15] reported that refugee and migrant children, children deprived of liberty, children living without parental care or proper shelter and children with disabilities are most vulnerable to COVID-19.

In terms of demographic characteristics, research conducted in [10] indicated that countries with high GDP per capita had an amplified number of reported severe COVID-19 cases and deaths. This may be due to more widespread testing in the developed countries, superior and transparent case reporting and better surveillance systems at national level. Frequent air-travel might be another possible cause of COVID-19 severity in the developed countries, as travel was identified as an important factor contributing to international viral spread [10]. For instance, [16] reported that the high numbers of COVID-19 cases in Jakarta, Indonesia, were caused due to high mobility of the people.

Likewise, smoking prevalence is also identified as being moderately to highly negatively correlated with COVID-19 infection rates. In [10], the authors surprisingly indicated that countries with a higher smoking rate, had lower frequency of critical cases and deaths. In addition, the authors reported a number of other possible predictors, which are associated with the total number of reported cases per million including: (i) days to lockdown (i.e. partial or full), (ii) commonness of obesity, (iii) median age of population, (iv) number of tests performed per million, and (v) days to the closure of borders. The study found a negative relationship between the total number of cases per million and the

number of days to lockdown, where a lengthier time preceding implementation of any lockdown, was linked with a lower number of detected COVID-19 cases per million. Furthermore, countries with high obesity rates among their population, higher median population age and longer number of days to border closure had considerably higher caseloads [10].

Studies have also indicated the average annual temperature of a country to be correlated with COVID-19 spread [16,17]. For instance, [17] found that the majority of the 10 000 new COVID-19 cases in the USA (10-day interval) are correlated with absolute humidity in a range of 4–6 g m$^{-3}$, and temperatures in a range of 4–11°C, thus concluding that low-temperature ranges are correlated with higher COVID-19 rates. Another research conducted in Brazil [18] found high solar radiation to be the main climatic factor that suppresses the spread of COVID-19. High temperatures and wind speed are also potential factors [18] correlated to COVID-19 spread. In summary, this work concluded that wind speed, temperature and increased solar radiation are the probable climatic factors that may steadily reduce the effects of the COVID-19 pandemic in Rio de Janeiro, Brazil.

Experimental results from [19] demonstrated that weather factors are more pertinent in predicting mortality rates in COVID-19 patients, when compared with other variables such as age, population and urbanization. The outcomes indicated that weather factors are more important as compared with age, population and urban percentage, while considering death rates due to COVID-19. A similar study in [20] proposed that humidity and temperature variations may represent significant factors affecting COVID-19 mortality rates.

Population density has also been reported as one of the relevant demographic attributes. Research outcomes in [21] indicated that in high population-density cities, it is difficult to enforce suitable distance between people coughing and sneezing. In turn, this may result in higher infection rates. Tsai & Wilson [22] stated that it is possible the disease will be transmitted to people facing homelessness. In the US, it is reported that more than 500 000 people were facing homelessness on any given night over the past decade (2007–2019). If cities enforce a lockdown to avoid COVID-19 transmission, it is unclear how and where homeless people will be relocated [22]. This can also be one of the potential causes of high infection and mortality rates in the US. Similar work in [23] reported that the high COVID-19 infection rates in Iran were positively correlated with population density and intra-provincial movement. Another study [24] investigated the morbidity and mortality rates of COVID-19 pandemic in various regions of Japan. The correlations between the morbidity, mortality rates and population density were found to be statistically significant while, lower morbidity and mortality rates were observed in regions with higher temperature and absolute humidity.

Researchers in [25] stated that the lockdown is an effective measure in limiting COVID-19 spread in densely populated areas. They also found that COVID-19 spread is negatively correlated with the latitude and altitude of the region. The study also found that there is no significant relationship between COVID-19 spread and population density, which contradict the findings of [23]. The study also suggested that strict lockdown procedures can effectively decrease the human-to-human infection propagation risk, even in densely populated regions, as stated in [21].

Air pollution, in [26], indicated negative correlation with COVID-19 infection rates. However, this contradicts the findings of [25]. Wu *et al.* [27] concluded that even a small increase (i.e. only 1 µg m$^{-3}$) in long-term exposure to PM2.5 results in a large increase (i.e. 8%) in COVID-19 mortality rates based on a US research study. By contrast, Zhu *et al.* [28] found a significant correlation between air pollution and COVID-19 infection rates. Positive correlations of PM2.5, PM10, CO, $NO_2$ and $O_3$ with confirmed COVID-19 cases were observed. However, the authors found $SO_2$ to be negatively associated with the number of daily confirmed cases of COVID-19. In [29], a direct relationship was discovered between air pollution and increased risk of hospital admission in Bangkok. It was also found that air pollution plays a significant role for the development of respiratory diseases such as pneumonia, asthma and chronic respiratory disease (CRD) leading to hospital admission. Elderly people are more fragile against the effect of air pollution and thus more vulnerable to respiratory diseases, similar to COVID-19. A similar study [30] also supported the argument presented in [29], indicating that exposure to air pollution could increase vulnerability and have negative effects on the prognosis of patients affected by COVID-19.

In addition to the aforementioned medical and demographic aspects of COVID-19, machine learning algorithms have been used in disease prediction and classification. For instance, Loey *et al.* [31] used a deep learning model and conventional machine learning methods for automated face mask detection. They deployed support vector machines, decision trees and ensemble method for the classification task. They claimed high accuracy results for both training and testing; however, the application of their system in the online context requires further details. Tuli *et al.* [32] used machine learning and mathematical models to detect the threat of COVID-19 around the globe. They claimed that their model

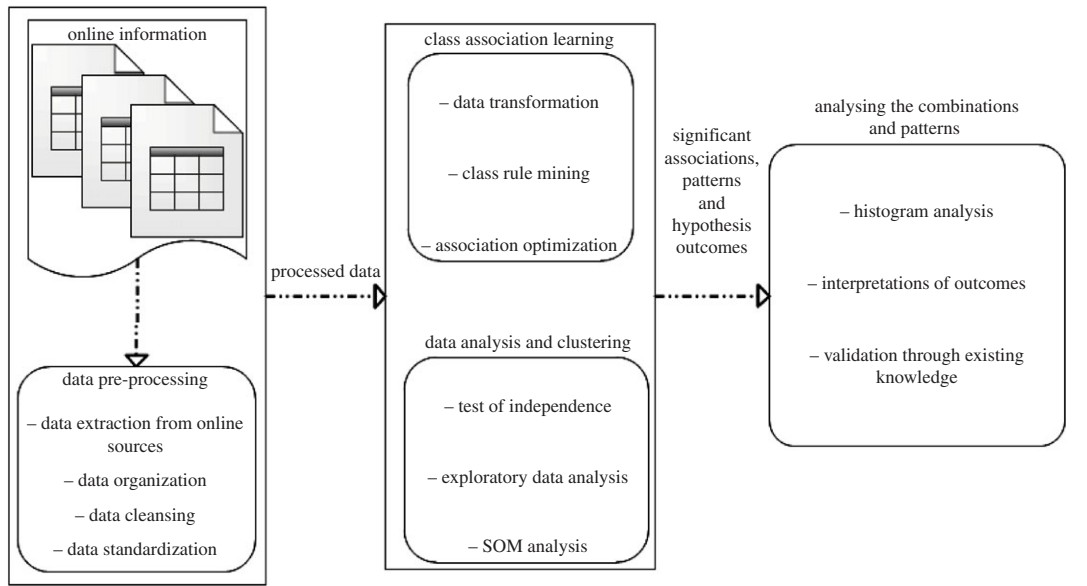

**Figure 1.** An overview of the modular framework for identification of significant demographic attributes, which are highly associated with COVID-19 death rates across the world.

outperforms the Gaussian model; however, there is a lack of benchmarking with other mathematical models. On the other hand, Yeşilkanat [33] used random forest to predict the future number of infected cases in 190 countries around the world, and compared the results with actual confirmed cases. RMSE values between 141.76 and 526.18 were reported; however, it would be interesting to show more results with other machine learning and statistical models for comparison purposes.

The aforementioned research studies investigated diverse aspects of COVID-19, specifically, association analysis and prediction using various medical and demographic attributes. However, the scope of these works is either limited to medical aspects or the analysis of individual association identification, where the outcomes indicated potential contradictions with other works. This might be due to several factors such as immature data/information about COVID-19 in the early stages, use of conventional statistical approaches and/or limitations in the combined analysis of multiple attributes which is presented in this study. More specifically, ongoing waves and variants of COVID-19 such as VOC 202012/01 further limit the generalization of existing similar studies conducted at earlier stages with immature data. We conduct a comprehensive analysis of complex associations and hidden patterns within the multi-dimensional data (gathered over a longer period of 1 year) while using the machine intelligence to investigate the impact of diverse demographic characteristics over the COVID-19 infection rate across the globe.

# 3. Material and methods

Combinations of intelligent algorithms are deployed to analyse the complex patterns and class associations between the multi-dimensional demographic attributes and COVID-19 death severity across different regions of the world. In the first step, the publicly available dataset is compiled from various sources (detailed in the following sections), comprising various demographic and COVID-19-related attributes across the globe. In the next step, data cleansing algorithms are used to remove outliers, where appropriate, and deal with missing records. The cleaned dataset is then normalized and passed on to pattern identification and association learning algorithms, to identify significant associations between the combined demographic attributes and the target attribute (i.e. death rate due to COVID-19). The statistical outcomes, associations and patterns are then fused together to draw the conclusions, while using existing information and experts' knowledge in the context of the underlying research question. Figure 1 summarizes the major building blocks for the proposed model, which are detailed in the following sections.

## 3.1. Dataset preparation

Exceedingly large and progressive data streams are publicly available, comprising numerous factors and statistics in relation to COVID-19. To investigate the research question set in this study, we used the

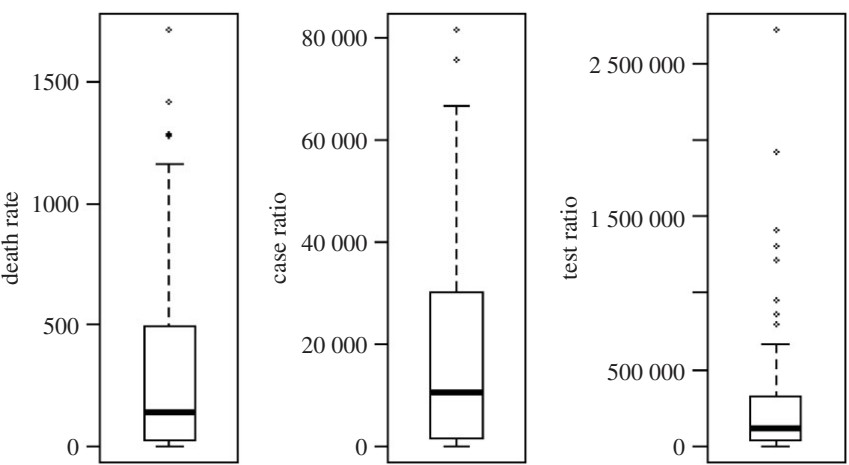

**Figure 2.** Boxplot visualization for the death rate (DpM), number of cases (CpM) and number of tests (TpM) across the globe.

publicly available dataset [34] until 8 January 2021 that comprises deaths per million population (DpM), cases per million population (CpM) and tests per million population (TpM) for each country across the globe (January 2020 to January 2021 in this study). Figure 2 shows the boxplot distributions for the selected attributes. Further explanation of the dataset, data capturing procedures and related ethical information is available in [34].

As previously discussed, several studies indicated significant relationships of certain demographic attributes with COVID-19, specifically, DpM. However, a detailed investigation is required to identify the complex associations and patterns within the multi-dimensional dataset. To investigate this hypothesis, we compiled several open-source demographic datasets [35–39], comprising a comprehensive list of 22 demographic attributes for countries around the globe, as shown in table 1. The parameters are selected based on recommendation from clinical domain experts as well as existing studies [6–30] as being potentially correlated with COVID-19 severity in different parts of the world. In total, there are 162 instances in the dataset each representing a single country.

The gathered data is then cleaned by eliminating outliers and missing values, specifically, countries containing some invalid entries (e.g. invalid or unnecessary names of countries such as 'Asian countries'), which were removed from the dataset. Figure 2 shows an example of outliers within the TpM attribute that were identified and eliminated using the box plot. Next, the cleaned numeric data were standardized using the z-score and forwarded to the pattern analysis and CARs algorithms.

## 3.2. Pattern analysis and rule mining

One of the major limitations associated with conventional statistical approaches is the inability to analyse complex patterns within a high-dimensional dataset. This study uses various demographic attributes (as listed in table 1) with diverse variation and ranges, which are difficult to be analysed by human experts or conventional statistical approaches, e.g. to draw conclusions from multiple combinations of different attributes. One effective means of dealing with multi-dimensional data visualization is SOM, an unsupervised form of artificial neural networks, performing a nonlinear projection of a high-dimensional space onto a lower-dimensional (typically, two-dimensional) map [40].

Topological properties of the input space are preserved in SOM, which use competitive learning, as compared with error minimization in supervised neural networks. The two-dimensional map representation is useful for pattern identification within the high-dimensional data such as the ones dealt with in this study. During the competitive learning phase, input data samples (e.g. a country's record in this study) are iteratively mapped to SOM, where a winning neuron (also called best matching unit) is identified based on the distance of its weights and the input vector. Weight update is performed within the specific neighbourhood radius resulting in similar samples being mapped closely together using

$$w_j(n+1) = w_j(n) + \eta(n)h_{ji(x)}(n)(x(n) - w_j(n)), \qquad (3.1)$$

where $\eta(n)$ is the learning rate and $h_{ji(x)}(n)$ is the neighbourhood function around the winner neuron $i(x)$. Both $\eta(n)$ and $h_{ji(x)}(n)$ are varied dynamically to achieve optimal outcomes. Further details, explanation and mathematical formulation of SOM can be found in [41].

**Table 1.** Demographic attributes names and description.

| attribute | description | attribute | description |
|---|---|---|---|
| Lung disease | death rate per 100 000 due to lung disease | Poverty ratio | poverty headcount ratio at $1.90 a day (% of population) |
| Hypertension | occurrence rate per 100 000 | Employment ratio | employment to population ratio, 15+ years, total (%) |
| Population density | people per square kilometre of land area | Smoking females | smoking prevalence, females (% of adults) |
| Female ratio | % of females in total population | Smoking males | smoking prevalence, males (% of adults) |
| Age_1 | population ages 0–14 (% of total population) | Air_pollution | PM2.5 air pollution, mean annual exposure ($\mu$g m$^{-3}$) |
| Age_2 | population ages 15–65 (% of total population) | Mortality rate_AP | mortality rate attributed to household and ambient air pollution, age-standardized (per 100 000 population) |
| Age_3 | population ages 65 and above (% of total population) | Mortality_Diab_CVD | mortality from CVD, cancer, diabetes or CRD between exact ages 30 and 70 (%) |
| Beds | hospital beds per 1000 people | Literacy rate | literacy rate, adult total (% of people ages 15 and above) |
| Forest Area | (% of land area) land area covered by forests | Physician | physicians per 1000 (include generalist and specialist medical practitioners) |
| Handwash | people with basic handwashing facilities including soap and water (% of population) | Health_Expenditure | current health expenditure (% of GDP) |
| Obesity | % of a country's obese population | Avg. Temperature | average yearly temperature (°C) |

As the attributes within the COVID-19 dataset (i.e. DpM, CpM, TpM) are in numerical form, we can use the SOM distance map to visualize these variables in the form of a two-dimensional plot. Figure 3 shows the mapping of countries over the SOM nodes, based on the distribution of three COVID-19 attributes, representing its severity levels. The algorithm automatically shapes the map (i.e. position of samples and nodes) using the distance metric between the codebook vectors of neurons/nodes. In other words, similar records (i.e. COVID-19 severity rates across the countries in this case) are mapped close to each other within the same node. Likewise, nodes with high similarity (i.e. nodes with smaller neighbouring distance) are positioned closely within the map, whereas dissimilar nodes are mapped far from each other. As an example, most of the severely affected countries (e.g. USA, UK, Spain, France, Belgium, Italy, etc.) are positioned within the left side and bottom-left nodes e.g. nodes 1–4, etc.) in figure 3, representing the similar behaviour of COVID-19 infection rates in these countries. On the other hand, the least affected countries (e.g. Thailand, Sri Lanka, Nepal, etc.) are placed in the top-right and right-side nodes (e.g. nodes 53–56, 60–64, etc.) within the map. This distribution clearly indicates the distinctive behaviour of COVID-19 severity levels across the globe, which requires further investigation in regard to its associations with other demographic characteristics, listed in table 1.

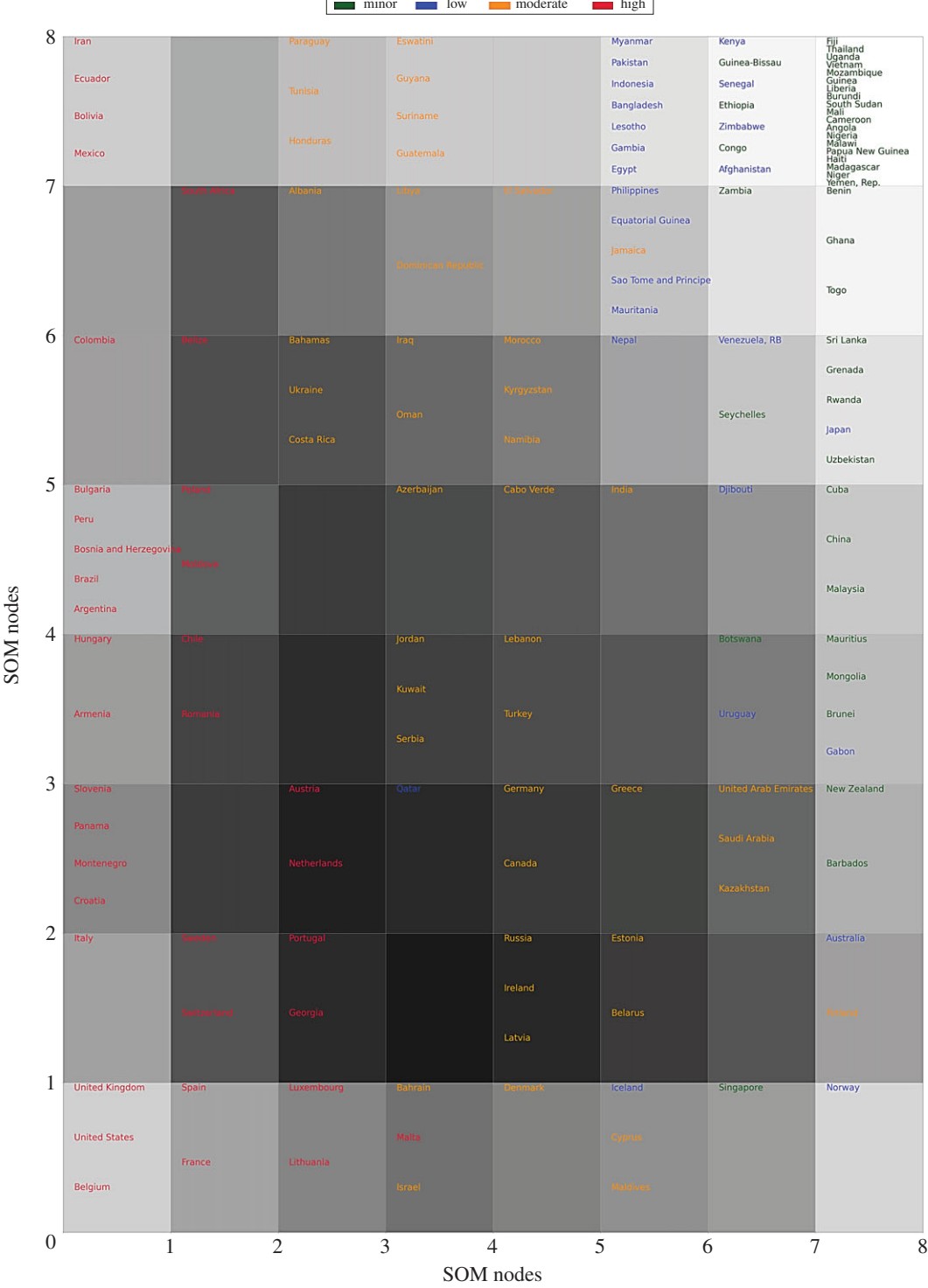

**Figure 3.** SOM distance map. The text colour of the country name indicates the 'death severity level' in the corresponding country, while a darker background represents higher neighbouring distance. The node positions start from bottom-left (node 1) and end at top-right (node 64).

In addition to the distance plot of figure 3, SOM provides heat maps, which are a powerful tool to visualize the individual behaviour of multiple attributes across the map. Figure 4 shows the distribution of individual factors across the SOM heat maps for all countries producing very useful visual information. For instance, 'Iceland' having 'low' DpM, is grouped together with high DpM countries (i.e. 'Cyprus and 'Maldives in node 6 of figure 3). However, mapping this information

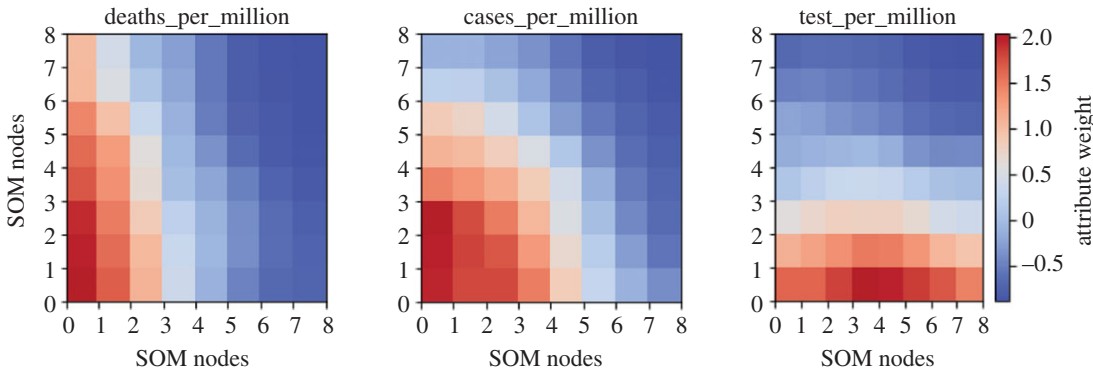

**Figure 4.** Heat maps representing the distribution of individual factors across the SOM map. The colour intensity (red to blue) indicates the magnitude of weight (high to low, respectively) associated with each attribute corresponding to neurons in the SOM map.

within the same node in figure 4 indicates that this grouping is due to the TpM in this zone, which is true in the case of Iceland (864 659/million in the dataset by January 2021). Secondly, the overlapping distributions observed for these attributes in figure 4 clearly indicate high positive correlations between these factors, which make sense in the case of the COVID-19 outbreak. For example, CpM is increasing with the increase in TpM in a country, which is also reported in related works [10,25].

While SOM produces powerful clustering and rich visual information within the numerical data, further investigation in relation to associations between multiple combinations of demographic characteristics and COVID-19 severity might be helpful to understand the complex patterns and inter-relationships within the categorical dataset. For this purpose, we use the special case of conventional rule mining known as CARs [42], where the consequent of a rule contains the target attribute (i.e. death severity in this case). As compared with conventional statistical techniques, CARs has the ability to identify frequently occurring patterns within a larger dataset that can be easily interpreted by humans in the form of rules. Let '$A$' be the attributes defined in table 1, containing $O = \{o_1, o_2, o_3, \dots o_N\}$ observations (i.e. countries' records) in the dataset, where each observation $o_i$ contains a subset of attributes $A$. The $X \rightarrow Y$ relationship in CARs indicates the disjoint item-set, i.e. $X \cap Y = \emptyset$, occurring in $O$, as *antecedents* and *consequents*, respectively. An important property of a rule is the corresponding *support count* ($\sigma$) representing the number of observations containing that item-set (i.e. attribute/s) which can be formulated as

$$\sigma(X) = |\{o_i | X \subseteq o_i, o_i \in O\}|. \tag{3.2}$$

The association of a rule ($X$) is usually controlled by *confidence* ($c$) and *support* ($s$) metrics, where

$$s\,(X \Rightarrow Y) = \frac{(\sigma(X \cup Y))}{N}. \tag{3.3}$$

In equation (3.3), $N$ represents the total number of countries in this study. The rule confidence measure $c$ is the percentage for which attribute $Y$ occurs with the presence of attribute $X$ and is represented as

$$c\,(X \Rightarrow Y) = \frac{(\sigma(X \cup Y))}{\sigma(X)}. \tag{3.4}$$

To account for the base popularity of both constituent items (i.e. $X$ and $Y$), a third measure called *lift* is used that measures the correlation between $X$ and $Y$ of a rule, indicating the effect of $X$ on $Y$, and is calculated as

$$lift(X \Rightarrow Y) = \frac{(s\,(X \cup Y))}{(s(X) * s(Y))} \tag{3.5}$$

A value of $lift(X \Rightarrow Y) = 1$ indicates independence between *antecedents* and *consequent*, whereas $lift(X \Rightarrow Y) > 1$ indicates positive dependence of $X$ and $Y$. A detailed explanation about CARs and the Apriori algorithm can be found in [42].

**Table 2.** Statistical metrics (i.e. quantiles) and clinical domain knowledge-based data transformation (numeric to categorical).

| attribute name | attribute categories | | |
| --- | --- | --- | --- |
| | low (L) | moderate (M) | high (H) |
| Lung Disease | Lung Disease $\leq$ 10 | 10 < Lung Disease $\leq$ 35 | Lung Disease > 35 |
| Hypertension | Hypertension $\leq$ 5 | 5 < Hypertension $\leq$ 19 | Hypertension > 19 |
| Population Density | PD $\leq$ 30 | 30 < PD $\leq$ 150 | PD > 150 |
| Female ratio | Female ratio $\leq$ 49 | 49 < Female ratio $\leq$ 51 | Female ratio > 51 |
| Age_1 | Age_1 $\leq$ 16 | 16 < Age_1 $\leq$ 38 | Age_1 > 38 |
| Age_2 | Age_2 $\leq$ 58 | 58 < Age_2 $\leq$ 68 | Age_2 > 68 |
| Age_3 | Age_3 $\leq$ 3 | 3 < Age_3 $\leq$ 15 | Age_3 > 15 |
| Beds | Beds $\leq$ 0.9 | 0.9 < Beds $\leq$ 4 | Beds > 4 |
| Air Pollution | Air Pollution $\leq$ 13 | 13 < Air Pollution $\leq$ 40 | Air Pollution > 40 |
| Mortality rate_AP | MAP $\leq$ 29 | 29 < MAP $\leq$ 145 | MAP > 145 |
| Poverty ratio | Poverty ratio $\leq$ 0.4 | 0.4 < Poverty ratio $\leq$ 20 | Poverty ratio > 20 |
| Employment ratio | Emp. ratio $\leq$ 50 | 50 < Emp. ratio $\leq$ 65 | Emp. ratio > 65 |
| Smoking males | Smoking $\leq$ 13 | 13 < Smoking $\leq$ 30 | Smoking > 30 |
| Smoking female | Smoking $\leq$ 1.5 | 1.5 < Smoking $\leq$ 12 | Smoking > 12 |
| Diabetes prevalence | Diabetes $\leq$ 5 | 5 < Diabetes $\leq$ 10 | Diabetes > 10 |
| Mortality (Diab_CVD) | Mortality_CVD $\leq$ 14 | 14 < Mortality_CVD $\leq$ 22 | Mortality_CVD > 22 |
| Literacy rate | Literacy rate $\leq$ 85 | 85 < Literacy rate $\leq$ 95 | Literacy rate > 95 |
| Physician ratio | Phys_rate $\leq$ 0.3 | 0.3 < Phys_rate $\leq$ 2.8 | Phys_rate > 2.8 |
| Health Expenditure | Health.Exped $\leq$ 4 | 4 < Health.Exped $\leq$ 8 | Health.Exped > 8 |
| Forest Area | Forest Area $\leq$ 10 | 10 < Forest Area $\leq$ 50 | Forest Area > 50 |
| Handwash | Handwash $\leq$ 30 | 30 < Handwash $\leq$ 95 | Handwash > 95 |
| Obesity | Obesity $\leq$ 8.5 | 8.5 < Obesity $\leq$ 25 | Obesity > 25 |
| Avg. Temperature | Avg. Temp $\leq$ 9 | 9 < Avg. Temp $\leq$ 25 | Avg. Temp > 25 |
| DpM (COVID-19) | *Minor*: DpM $\leq$ 25, *low*: 25–100, *moderate*: 100–500, *high*: DpM > 500 | | |
| CpM (COVID-19) | *Minor*: CpM $\leq$ 1200, *low*: 1200–4600, *moderate*: 4600–35 K, *high*: CpM > 35 K | | |
| TpM (COVID-19) | *Minor*: TpM $\leq$ 15 K, *low*: 15–36 K, *moderate*: 36–200 K, *high*: TpM > 200 K | | |

To deploy CARs in this study, the numeric dataset was transformed into categorical form using statistical information (i.e. quantiles and interquantile ranges), as well as expert knowledge, where appropriate. Table 2 summarizes the multi-scale categories for the demographic attributes as transformed using statistical metrics and histogram distributions. Similarly, the COVID-19 attributes (e.g. DpM) are also categorized as 'Mild' to 'Severe', indicating lowest to highest severity levels, respectively, across the globe. The final categorical data representations contain uniform representations of all attributes forming the knowledge base for CARs to learn the associations between combinations of multiple attributes and the target DpM in the COVID-19 dataset.

Figure 5 shows the frequency distributions of the categorical attributes within the transformed dataset. It can be noted that all attributes have uniform categories as low (L), medium (M) and high (H) with the additional category of minor (Min) for the COVID-19 attributes.

The histograms demonstrate normalized distributions for the demographic attributes indicating categorization of the numerical dataset. To find the individual relationships between the DpM and demographic attributes, we initially deployed the $\chi^2$-test, which is one of the most commonly used statistical tests of independence for categorical data. The $\chi^2$-test between the DpM and CpM provided $\chi^2$ = 162.19 with a *p*-value of $2.2 \times 10^{-16} \ll 0.05$, clearly indicating the rejection of the *null hypothesis*, thus concluding that DpM is highly dependent on CpM in a country that aligns with the existing study [10] as well as SOM-based analysis (figure 4). Similarly, the $\chi^2$-test between TpM and CpM produced $\chi^2 = 69.46$ with a *p-value* of $1.942 \times 10^{-11} \ll 0.05$, also indicating the rejection of the *null*

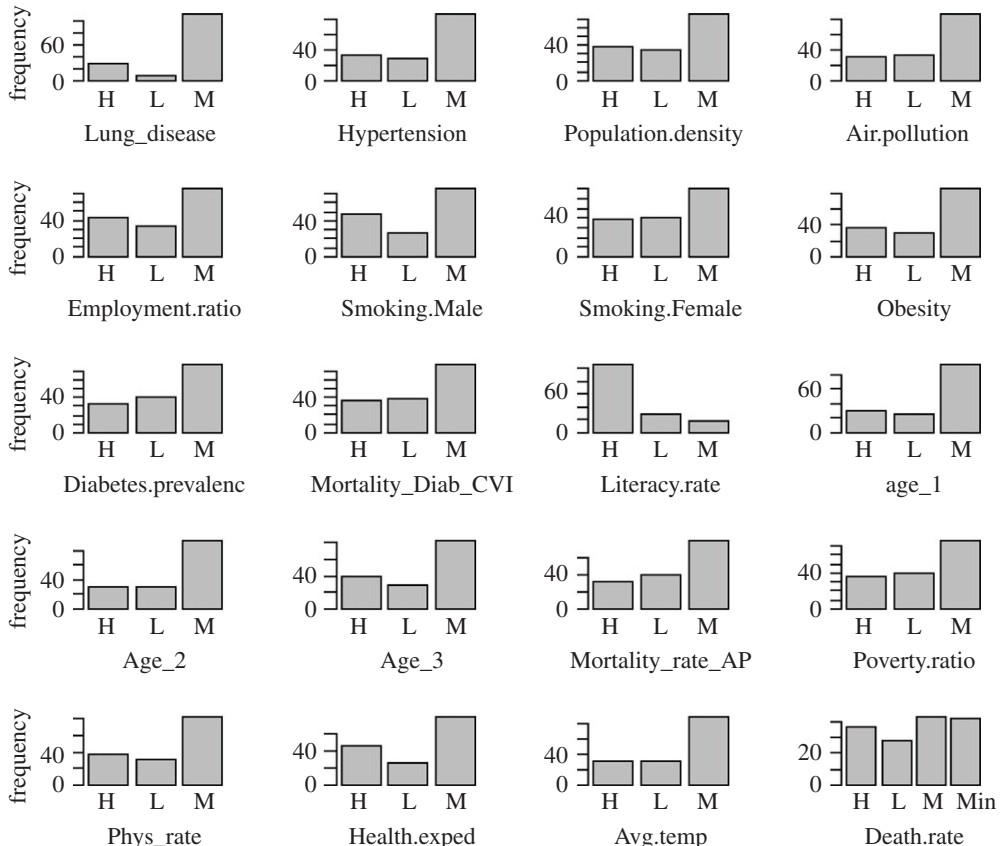

**Figure 5.** Distribution of the categorized attributes (table 2) within the dataset representing the world demographics and COVID-19 severity. High (H), low (L), moderate (M) and minor (Min).

*hypothesis*. These findings align with the SOM-based outcomes of figures 3 and 4, implying that a higher number of tests will produce a high number of cases, which will, ultimately, result in a high DpM for the corresponding country.

Table 3 indicates the outcomes from the $\chi^2$-test of independence between each demographic attribute and DpM. It can be observed that the value of $\chi^2$ for certain individual attributes, e.g. Age, Poverty_ratio, Obesity, Avg_temperature and Female_smokers, indicates high dependence with the target attribute of DpM (i.e. $p$-value $\ll 0.05$). However, it is important to investigate the combined associations of these attributes with the varying nature of COVID-19 severity across the globe.

As mentioned earlier, CARs can produce the desired associations while using the categorized demographic data (i.e. table 2) as antecedents and DpM as the consequent in the rules. One of the limitations of conventional rule mining is the generation of a high number of rules, which make them impractical for interpretation by traditional approaches or human experts. However, this issue can be resolved using sequential filtration of irrelevant rules with varying threshold values for parameters $c$ and $s$. The selection of optimal values for these thresholds entirely depends upon the nature of the problem and the data itself [43]. Based on empirical experiments, we performed rule filtration while optimizing several parameters, which included confidence (minimum confidence = 0.9), minimum length = 2, maximum length = 5, thus resulting in the extraction of a compact list of highly associated rules. As per the research question in this study, we extracted conditional rules based on DpM severity levels (i.e. minor, low, moderate and high) as consequent, which further limits the generation of a larger set of rules. In addition, we used redundant rules elimination [44] to filter out repetitive rules and, therefore, resulting in the list of the most representative ones.

# 4. Results and discussion

In order to investigate the potential patterns within the dataset and class associations between the demographic attributes and COVID-19 severity, specifically, DpM around the world, experiments were

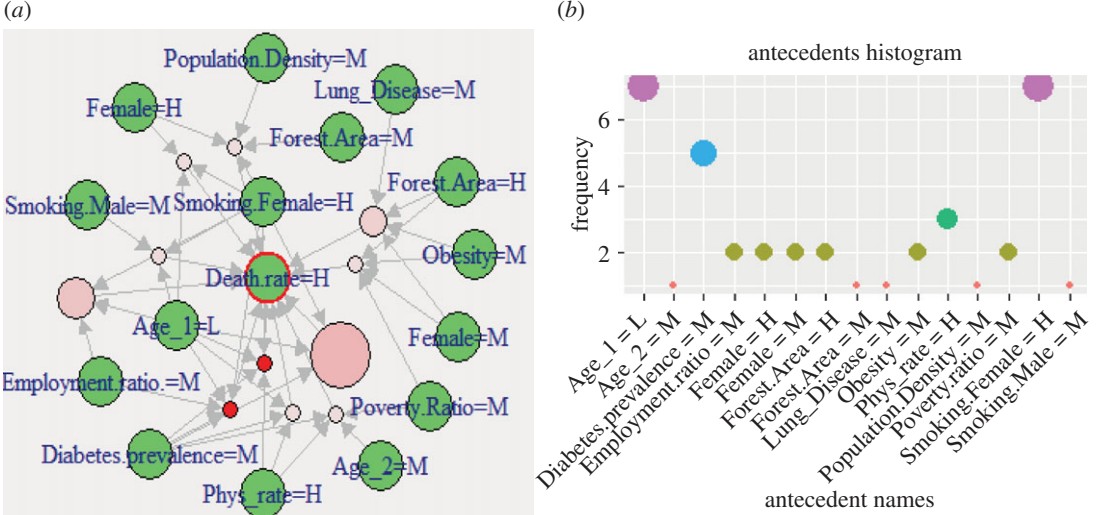

**Figure 6.** (a) Visualization of representative rules (red circles) between multiple demographic attributes and *high DpM* (green circles). A larger-sized red circle indicates higher lift value for that rule and vice versa. (b) Antecedents' histogram within CARs (i.e. demographic attributes' frequency) for the *high DpM* shown in (a).

**Table 3.** $\chi^2$-test of independence between the demographic attributes and DpM (d.f. = 6).

| attribute name | $\chi^2$ | $p$-value | attribute name | $\chi^2$ | $p$-value |
|---|---|---|---|---|---|
| Lung disease | 13.04 | 0.041 | Smoking females | 22.53 | 0.0009 |
| Hypertension | 22.83 | 0.0008 | Smoking males | 4.86 | 0.56 |
| Population density | 3.58 | 0.73 | Diabetes prevalence | 6.92 | 0.32 |
| Female ratio | 8.92 | 0.17 | Mortality (Diab_CVD) | 26.25 | 0.00019 |
| Age 0–14 (Age_1) | 32.87 | $1.16 \times 10^{-5}$ | Literacy rate | 31.46 | $2.06 \times 10^{-5}$ |
| Age 15–65 (Age_2) | 29.7 | $3.97 \times 10^{-5}$ | Physician per 1000 | 32.72 | $1.18 \times 10^{-5}$ |
| Age 65+ (Age_3) | 23.11 | 0.0004 | Health expenditure | 33.41 | $8.72 \times 10^{-6}$ |
| Beds.per.1000 | 25.81 | 0.0002 | Forest area | 1.71 | 0.94 |
| Air Pollution | 15.58 | 0.016 | Basic handwash | 33.43 | $8.63 \times 10^{-6}$ |
| Mortality rate_AP | 37.75 | $1.25 \times 10^{-6}$ | Obesity | 41.29 | $2.53 \times 10^{-7}$ |
| Poverty ratio | 39.42 | $5.90 \times 10^{-7}$ | Avg. temperature | 28.30 | $8.24 \times 10^{-5}$ |
| Employment ratio | 16.19 | 0.011 | | | |

conducted using both numeric and categorical representations of the dataset, comprising the demographic and COVID-19-related attributes in tables 1 and 2. The CARs algorithm was used with the parametric configurations and rule filtration explained in §3.2, while considering the listed attributes as *antecedents* and target DpM as a *consequent* of CARs. The specific objective of these experiments was to analyse the associations between the extreme levels of DpM (i.e. severe and mild) and certain demographic attributes, particularly, the ones associated with health, environmental and economic indicators of a country.

Figure 6a demonstrates the 11 representative rules (shown as light-pink and red circles) comprising the list of attributes (green circles) identified as highly associated with the high DpM across the globe. The size and colour intensity (i.e. red colour) of the circles relates to the relative strength of the rule in terms of *confidence* and *lift* measures, respectively. These non-redundant rules indicate significant associations (with *confidence* > 0.9 and *lift* ≥ 3.64) between the DpM and multiple demographic attributes such as *high* values for (i) Phys_rate, (ii) smoking_females, *moderate* levels of (iii) obesity, (iv) population density, (v) diabetes_prevalence, (vi) poverty_ratio, while *low* categories of (vii) younger population (i.e. Age_1).

**Table 4.** Antecedents in class-rules with high association between demographic attributes and *high death-rate* (consequent), *lift* > 3.64, *confidence* > 0.9, *support* > 0.065.

| |
|---|
| Diabetes.prevalence = M, Age_1 = L, Phys_rate = H |
| Smoking.Female = H, Female = H, Age_1 = L |
| Employment.ratio. = M, Smoking.Female = H, Age_1 = L |
| Smoking.Male = M, Smoking.Female = H, Age_1 = L |
| Smoking.Female = H, Diabetes.prevalence = M, Age_1 = L |
| Smoking.Female = H, Diabetes.prevalence = M, Age_1 = L, Phys_rate = H |
| Employment.ratio. = M, Smoking.Female = H, Diabetes.prevalence = M, Age_1 = L |
| Obesity = M, Forest.Area = H, Female = M, Poverty.Ratio = M |
| Lung_Disease = M, Obesity = M, Forest.Area = H, Female = M |
| Diabetes.prevalence = M, Age_2=M, Poverty.Ratio = M, Phys_rate = H |
| Population.Density = M, Smoking.Female = H, Forest.Area = M, Female = H |

Despite the elimination of redundant rules and restricted parametric constraints (e.g. *lift*, *confidence*), individual occurrences of different attributes within the representative rules might be helpful for visual analysis. For this purpose, we extracted the frequency histograms within the antecedents of rules as shown in figure 6*b*. Frequency histograms help to visualize a more complex and larger set of rules to further investigate the significance of individual factors within the list of representative rules. However, it is important to consider the associations, when antecedents are combined with other factors (i.e. how the association varies with varying combinations in *antecedents*) as shown in figure 6*a*.

Table 4 further shows the list of *antecedents* for the rules presented in figure 6*a*. These outcomes clearly indicate the significant associations between a *high* DpM and certain demographic attributes, specifically, *low* poverty_ratio and young population, while *high* female_smoking and medical facilities (e.g. Phys_rate). The outcomes align with existing research, for instance [10], that also reported significant association between the economic (GDP) condition of a country and COVID-19 spread. However, the results of our work also consider the impact of combined attributes (e.g. smoking_female = H appears with the *low* Age_1 and *high* Phys_rate), which is an important aspect to be further analysed.

Table 4 and figure 6 demonstrate that attribute smoking_female is highly associated with high levels of DpM. Recent studies [6,9] reported a positive correlation between smoking_prevalence and COVID-19 deaths, which aligns with our outcomes. On the other hand, research conducted in [10] reported contradictory outcomes, indicating negative correlation of smoking prevalence and COVID-19 impacts. We used the gender information (i.e. male versus female) in combination with the smoking ratio (male, female), which helps to further investigate contradicting outcomes in previous studies [6,9,10], while measuring the combined relationship. Our findings demonstrate that countries with a higher ratio of female smokers are affected more as compared with that of countries with more male smokers. Likewise, [6,11] reported males being at more risk than females; however, when smoking ratio is combined together with the gender attribute in our research, it produces contradicting outcomes. This can be further validated with figure 3 (i.e. SOM map for global distribution of DpM), where most of the countries containing high female smokers (e.g. UK, Spain, Chile, Montenegro, France, USA, Luxembourg, Bosnia and Herzegovina, etc.) overlap with countries appearing within the high DpM area of the SOM map (i.e. left/bottom-left nodes). The outcome also aligns with a fact sheet [45] issued by the CDC, which states that smoking damages the human immune system and can make the body more vulnerable against COVID-19 attacks.

Furthermore, in most of the rules shown in table 4, countries with *low-to-moderate* poverty_ratio indicated significant associations with *high* DpM, which puts credence on the existing findings [10]. This factor can also be validated using the SOM map distribution (figure 3), indicating *high* DpM in most of the *high* GDP countries (i.e. bottom-left nodes). The significant association between the poverty_ratio and DpM may also be due to several factors such as the limited availability of medical resources in low GDP countries, less travelling (i.e. national and international) due to limited GDP and therefore causing less spread of COVID-19, effective lockdown policy and less tourism. Furthermore, the limited number of COVID-19 tests (i.e. low TpM) carried out in low GDP countries

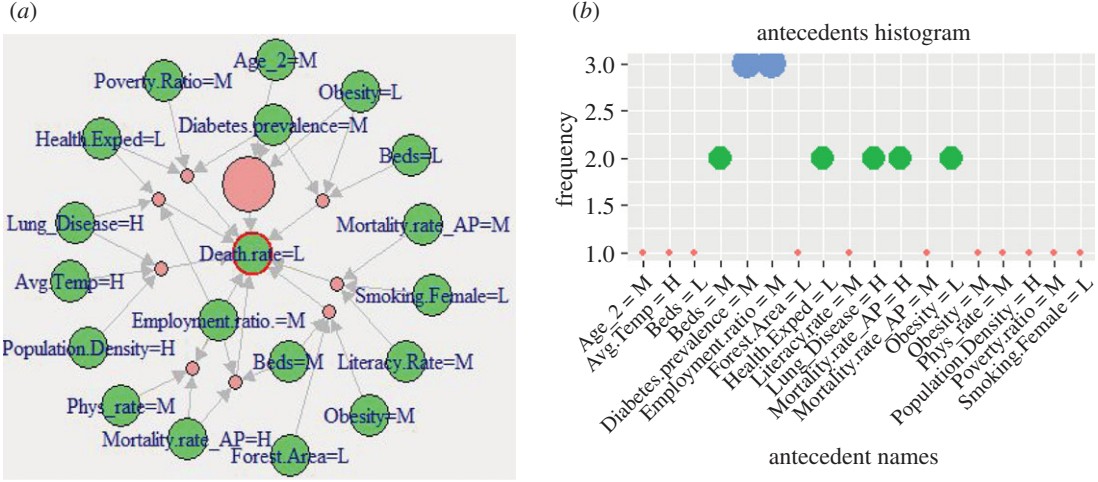

**Figure 7.** (a) Visualization of representative associations between multiple attributes and *low death rate*. (b) Antecedents' histogram in CARs for the *low* DpM shown in (a).

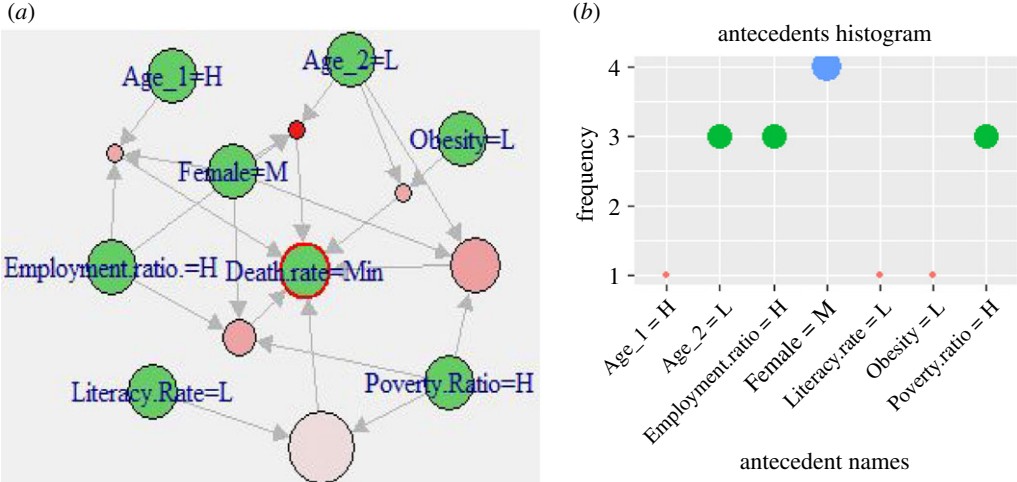

**Figure 8.** (a) Visualization of representative associations between multiple attributes and *minor death rate*. (b) Antecedents histogram in CARs for *minor* DpM shown in (a).

also significantly reduces DpM as can be seen in figure 4, where DpM is high positively correlated to TpM.

These outcomes demonstrate the significance of combined attribute analysis, producing more reliable outcomes and comprehensive insight of inter-relationships. On the other hand, most of the existing studies are reporting on these attributes individually while using immature datasets, and are therefore insufficient in drawing general conclusions about the diversity in DpM distribution across the globe.

In a similar way, figures 7 and 8 show the rules comprising the demographic attributes that indicate significant associations with the *low* and *minor* DpM levels, respectively. The corresponding *antecedents* are reported in tables 5 and 6, respectively, which clearly indicate frequent occurrences of *low* Obesity, Age_2, and *high* Poverty_ratio, Avg_temperature and Employment_ratio. More specifically, Age_2 appeared as *low* here as compared with Age_1, which is comparatively *high* which means that countries with a high ratio of aged population are affected more by COVID-19, compared with those with a high ratio of younger population. This also aligns with existing findings, such as [12–14] and WHO reports [15] indicating younger people and, specifically, children are less affected. An example scenario in our findings is Pakistan (with a low ratio of aged population, Age_3: 4.3%) versus the United Kingdom (with higher ratio of aged people: Age_3: 18.5%). This outcome also aligns with the SOM heat map (figure 3), where Pakistan appears in low affected areas in the map (i.e. top-right nodes), while the UK appears in the bottom-left (i.e. severely affected) areas of the map.

**Table 5.** Antecedents in class-rules with high association between selected factors and *low death-rate* (consequent), *lift* > 3.5.

| |
|---|
| Smoking.Female = L, Literacy.Rate = M, Mortality.rate_AP = M |
| Lung_Disease = H, Employment.ratio.=M, Health.Exped = L |
| Diabetes.prevalence = M, Poverty.Ratio = M, Health.Exped = L |
| Obesity = L, Diabetes.prevalence = M, Beds = L |
| Lung_Disease = H, Population.Density = H, Avg.Temp = H |
| Obesity = L, Diabetes.prevalence = M, Age_2 = M |
| Employment.ratio. = M, Mortality.rate_AP = H, Phys_rate = M |
| Employment.ratio. = M, Beds = M, Mortality.rate_AP = H |
| Obesity = M, Forest.Area = L, Beds = M |

**Table 6.** Antecedents in class-rules with high association between selected factors and *minor death-rate* (consequent), *lift* > 3.42, *confidence* > 0.9.

| |
|---|
| Literacy.Rate = L, Poverty.Ratio = H |
| Obesity = L, Age_2 = L, Poverty.Ratio = H |
| Female = M, Age_2 = L, Poverty.Ratio = H |
| Employment.ratio. = H, Female = M, Age_2 = L |
| Employment.ratio. = H, Female = M, Age_1 = H |
| Employment.ratio. = H, Female = M, Poverty.Ratio = H |

Table 6 indicates an important aspect of significant association between the obesity level and DpM severity. The *antecedents*' histogram (figure 8b) indicates that *low* obesity is highly associated with *minor* DpM, whereas obesity is *moderate* when DpM is *high* as shown in table 4 and figure 6. This implies that DpM increases with an increasing obesity population ratio, which is consistent with the results of a recent COVID-19 study [7], which shows a positive correlation between COVID-19 infections and obesity. However, it is important to note that our results indicate that a *low* obesity appears in combination with *high* Poverty_ratio and *low* Age_2, which indirectly represents *lower* GDP countries. In other words, the combination demonstrates strong associations between these attributes and *minor* DpM, while simultaneously, the inter-relationship between these attributes. According to [46], most of the *low* GDP countries (i.e. *high* poverty ratio in our study) are reported with a *high* global hunger index (GHI), which indirectly validates the combined appearance of *low* obesity and *high* poverty ratio in the case of *minor* DpM. Furthermore, these outcomes clearly indicate that the obesity attribute reported in existing works, such as [7], is highly dependent upon other demographic characteristics of a region that might alter the outcomes, when analysed in combination with these demographic attributes.

In summary, the association outcomes in tables 4–6 indicate that certain demographic attributes, specifically, Obesity level, Poverty ratio, Age group, Annual temperature and Smoking prevalence, combined with gender information (i.e. smoking_females, males), are highly associated with COVID-19 severity levels (i.e. DpM) across different countries. Likewise, several demographic attributes related to medical facilities (e.g. Health_expenditure, Physician_ratio and Beds availability), environmental attributes (e.g. Forest area, Handwash facilities, etc.) and economic factors (e.g. Poverty ratio, Employment ratio, etc.) indicated comparatively partial associations with the COVID-19 severity distribution across the globe.

As the demographic and COVID-19 dataset are primarily in numerical form, we can use the SOM heat maps to visualize the distribution of all demographic attributes in a two-dimensional plot as shown in figures 9 and 10. This also helps to simultaneously visualize the inter-relationships between these attributes. For instance, Test_ratio and Case_ratio in figure 9 show similar patterns across the map indicating a high correlation between them. Interestingly, the outcomes in figure 9 align with the CARs results (tables 4–6), indicating the significant dependence between DpM and the certain demographic attributes across the world. For instance, heat maps for the Age_1 distribution in

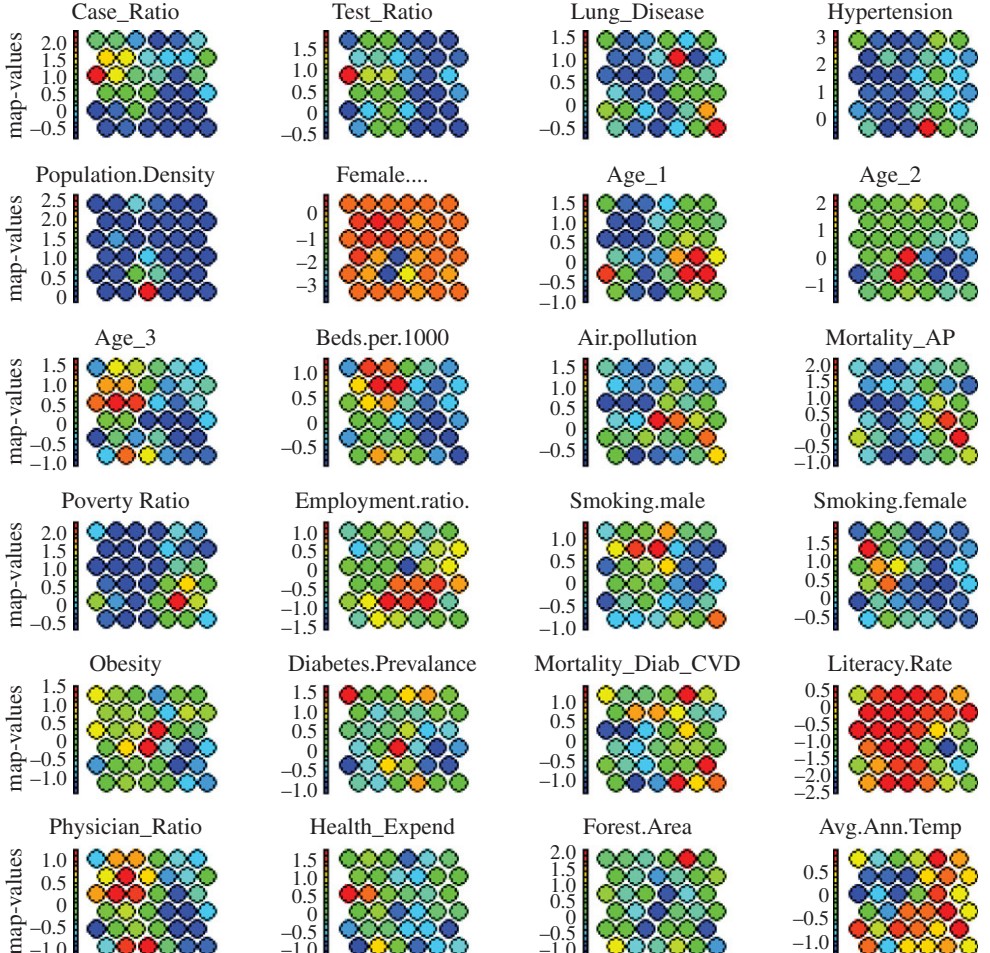

**Figure 9.** Demographic attributes' distributions across the SOM heat map. Colour intensity (blue to red) indicates the magnitude (low to high, respectively) of the weight associated with each attribute corresponding to neurons in the SOM map.

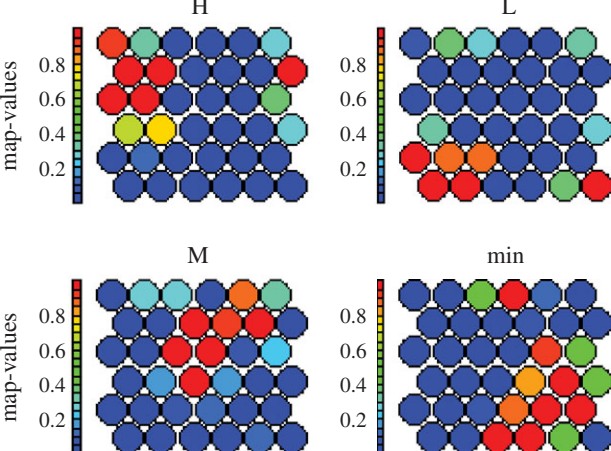

**Figure 10.** DpM distributions across the SOM heat map. Colour intensity (blue to red) indicates the magnitude (low to high, respectively) of the weight associated with each DpM level corresponding to neurons in the SOM map.

figure 9 inversely correlate with the heat map distribution of DpM in figure 10, indicating *higher* Age_1 (e.g. nodes 10, 11, 16–18 in figure 9), *lower* DpM (nodes 10, 11, 16–18 in figure 10) and vice versa, which is similar to existing findings [9,12–14]. Similarly, Obesity and Female_smoking levels indicate direct correlations with DpM, which also aligns with the CARs outcomes (i.e. categorical data). Likewise, the demographic attributes related to medical facilities, environment and economic indicators also

indicated relationships with DpM similar to CARs associations. Furthermore, the inter-relationships identified between Age, Obesity and Poverty_ratio in CARs (tables 4–6), are also produced by the SOM heat maps in figure 9.

It is important to consider the ongoing and dynamic nature of COVID-19 severity levels across the globe. More specifically, the continuing COVID-19 waves and variants such as VOC 202012/01 might affect the generalization and outcomes of the existing predictive approaches. Typical examples may include the temperature and population density of a country. Some of the existing studies [17–20,22] reported an inverse relationship between the temperature of a country and COVID-19 spread. We carried out SOM analysis to investigate this further, which, interestingly, indicated a moderate negative relationship between the temperature of a country and the corresponding DpM, which is also reported in CARs outcomes (see table 5 and figure 7). Likewise, the $\chi^2$-test of dependence produced a $p$-value of $8.24 \times 10^{-5} \ll 0.05$, indicating significance dependence between the DpM and average temperature of a country. These statistics and SOM outcomes indicate at least a partial relationship between the DpM and average annual temperature of a country, which also places credence on the aforementioned studies. However, as mentioned earlier, the outcomes might vary when analysed in combination with other demographic characteristics and geographical locations. For example, USA, Iraq and India, with comparatively high annual average temperatures, are listed as severely affected countries, which contradicts the above argument. This implies that while the argument is true in most of the cases, the results in this work as well as reported in the existing studies, are insufficient to draw general conclusions about the interdependence of COVID-19 severity and annual temperature of a country, specifically in the given circumstances of ongoing waves, variants and dynamic spread of COVID-19 across the globe.

Population density on the other hand, has been considered an important but contradictory factor in existing works. For instance, [23,26] reported dense population areas being positively correlated with COVID-19 cases in contrast to [25], which reported that correlation is not significant. The outcomes from SOM and CARs in proposed work indicated that population density is irrelevant, which agrees with the research outcomes reported in [25]. This indicates that COVID-19 spread in high-density population regions can be controlled with the effective management, specifically, lockdown policy implementation as reported by the WHO and [21,25].

Finally, the Air-pollution indicated high negative association with DpM (figures 9 and 10), which aligns with the outcomes reported in [25,28]. However, the outcomes contradict the findings reported in [27,29,30]. Likewise, Hypertension and Lung_disease in figures 9 and 10 show moderate negative relationship with DpM, which aligns with CARs outcomes (figure 6), but contradicts the findings in [6,9,47]. There might be several factors for this contradiction, specifically, (i) the use of conventional statistical analysis of individual associations in existing studies, (ii) the nature of this study, which is based on demographic attributes and not COVID-19 health-related symptoms, and (iii) the use of a premature dataset about COVID-19 infections in existing works, which may produce variations in results at later stages of the disease.

# 5. Conclusion and future directions

This research proposed a framework of data analytics algorithms to investigate which demographic characteristics are highly associated with severe death rates due to COVID-19 in different countries. The study performed a comprehensive analysis using well-established clustering and class rule mining algorithms to investigate COVID-19 death-rate associations with multiple individual and combinations of demographic attributes. Our results demonstrate that certain demographic attributes, specifically, age distribution, poverty ratio, female smokers percentage, obesity level and average annual temperature of a country, are significantly associated with COVID-19 death rate distribution. This is potentially an important finding, implying that various demographic attributes can be used as markers to identify COVID-19 spread and severity levels, leading to various aspects (e.g. social, economic, cultural, healthcare, educational, etc.) and a bunch of other related conclusions, which may be helpful to policy makers, health professionals and individuals for the effective management and control of the disease.

The authors believe that the complex associations and patterns within the multi-dimensional demographic attributes in this work are more comprehensively studied, when compared with the use of classical statistical approaches, reported in most of the existing works. Our findings demonstrate that certain individual attributes (e.g. age, gender, GDP ratio), when combined with other

demographic characteristics (e.g. smoking ratio, obesity), produce varying outcomes in contrast to some of the existing works that use conventional statistical techniques with inability to explore the complex patterns within the high-dimensional data. It is also vital to consider the dynamic and ongoing nature of the COVID-19 spread across the globe that might affect the conclusions made in some of the existing studies using insufficient data at premature stages of the disease. As an example, India had fewer (i.e. 25% only) cases in the first five months of the outbreak (i.e. February to June 2020), however, within the following two months only (July and August 2020), 75% of the total number of cases appeared. This type of dynamic CpM might influence the outcomes and generalization of existing works carried out with an insufficient dataset at earlier stages. Finally, we identified several attributes, including hypertension, lung disease, mortality rate (CVD and diabetes) and medical facilities (e.g. beds, physician rate, etc.), which are also partially associated with COVID-19 spread and may set a baseline for future investigations.

Ethics. No ethical approval was required for this study.

Data accessibility. COVID-19 infection data (deaths, cases and tests per million population for each country): Worldometer, Reported Cases and Deaths by Country, Territory or Conveyance, https://www.worldometers.info/coronavirus/#countries.

Global demographic attributes: World Bank open data, https://data.worldbank.org.

Hypertension and lung disease: Worldlifeexpectancy, https://www.worldlifeexpectancy.com/cause-of-death/hypertension/by-country/.

Smoking ratio: The Tobacco Atlas, https://tobaccoatlas.org/.

Obesity level: Central Intelligence Agency (The world factbook), https://www.cia.gov/the-world-factbook/field/obesity-adult-prevalence-rate.

Annual Avg. temperature: WayBackMachine, https://web.archive.org/web/20150905135247/http://lebanese-economy-forum.com/wdi-gdf-advanced-data-display/show/EN-CLC-AVRT-C/.

Furthermore, the datasets supporting this article have been uploaded as part of the electronic supplementary material.

Authors' contributions. Conceptualization, W.K. and A.H.; design, M.A., W.K., A.H.; methodology, W.K., A.H., P.L. and R.N.; software, W.K.; S.A.K.; R.N.; validation, W.K., A.H., M.A.; dataset collection, W.K., S.A.K.; writing the original draft, W.K., S.A.K., A.H., P.L.; editing, P.L., R.N.

Competing interests. Authors have no competing interests.

Funding. Authors have no funding to report for this research.

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
