## [Reviewer comments · Royal Society Open Science]

Review History

RSOS-201823.R0 (Original submission)

Review form: Reviewer 1

Is the manuscript scientifically sound in its present form?

Yes

Are the interpretations and conclusions justified by the results?

Yes

Is the language acceptable?

Yes

Do you have any ethical concerns with this paper?

No

Have you any concerns about statistical analyses in this paper?

No

Recommendation?

Accept as is

Comments to the Author(s)

- I found the paper well written and technically sound in a timely topic. I enjoyed reading it. I recommend its acceptance.

Review form: Reviewer 2**Is the manuscript scientifically sound in its present form?**

Yes

Are the interpretations and conclusions justified by the results?

Yes

Is the language acceptable?

Yes

Do you have any ethical concerns with this paper?

No

Have you any concerns about statistical analyses in this paper?

No

Recommendation?

Accept with minor revision (please list in comments)

Comments to the Author(s)

The study raises interesting questions, in particular understanding the covid-19 infection severity and whether it is associated with one or more demographic feature/s. The work is certainly timely, and address an issue that is challenging and 'perhaps' overlooked in the research community.

The paper is well written and structured, the literature review is thorough, and up to date with an interesting critical discussion of various relevant recent studies with some contradicting research reports and findings in the area of covid19 data-related work. The methods used are sound and well presented including good use of illustrations and presentation of tabulated results.

So overall, I think this is a good and timely study, and certainly can be accepted for publications as I believe the research community would benefit from it. That said, I have the following suggestions/ questions:

1. I would have liked to see more discussion of the dataset used? It is stated that the authors used 22 attributes? First, why these 22 only, and second what is the total number of instances in the dataset (did you consider all countries)? I suggest you made this clear at the beginning of the materials section where you discuss the dataset
2. Figure 3 is difficult to read, I wonder if there is a better way to present the results

3. The authors utilised class-association rules and self-organising maps to uncover these 'potential' associations. Is there a reason for this? Why not other machine learning techniques? I suggest just give some justification for these choices

Review form: Reviewer 3

Is the manuscript scientifically sound in its present form?

Yes

Are the interpretations and conclusions justified by the results?

Yes

Is the language acceptable?

Yes

Do you have any ethical concerns with this paper?

No

Have you any concerns about statistical analyses in this paper?

No

Recommendation?

Accept with minor revision (please list in comments)

Comments to the Author(s)

The article presents an analysis of the effect of COVID-19 by considering various demographic characteristics. The paper is well-written, but it would probably help readability if a list of the main abbreviations was included in the introduction.

There is an excessive number of percentages in the second paragraph of the introduction. These could be replaced by qualitative observations that explain the timeliness of the present study.

The methodologies presented are sound and adequately explained. However, a wide range of other methodologies could have been alternatively used. For this reason, it would make sense to include in the text some additional explanations regarding the reasons that made the authors choose these specific methodologies.

The authors use publicly available data, collected up to a specific date. It may be worthwhile to consider including additional (recent) data in order to test the consistency of the followed methodological approach and ensure the relevance of the conclusions over time.

The text could become clearer if some observations were expressed in more qualitative terms. For example, instead of stating that "A has negative/positive association with B", one could just state that "when A is higher/lower, B seems to be lower/higher".

Some additional efforts could be made to explain contradictions between the conclusions reached in this present work and conclusions reached in prior works.

Decision letter (RSOS-201823.R0)

Dear Dr Khan

On behalf of the Editors, we are pleased to inform you that your Manuscript RSOS-201823 "Analysing the Impact of Global Demographic Characteristics over the COVID-19 Spread Using Class Rule Mining and Pattern Matching" has been accepted for publication in Royal Society Open Science subject to minor revision in accordance with the referees' reports. Please find the referees' comments along with any feedback from the Editors below my signature.

Please submit your revised manuscript and required files (see below) no later than 7 days from today's (ie 07-Jan-2021) date. Note: the ScholarOne system will 'lock' if submission of the revision is attempted 7 or more days after the deadline. If you do not think you will be able to meet this deadline please contact the editorial office immediately.

on behalf of Prof Marta Kwiatkowska (Subject Editor)
openscience@royalsociety.org

Associate Editor Comments to Author:

Three reviewers have offered a number of minor recommendations to improve your paper - final acceptance will be contingent on your incorporating these changes into the paper.

Reviewer comments to Author:

Reviewer: 1

Comments to the Author(s)

- I found the paper well written and technically sound in a timely topic. I enjoyed reading it. I recommend its acceptance.

Reviewer: 2

Comments to the Author(s)

The study raises interesting questions, in particular understanding the covid-19 infection severity and whether it is associated with one or more demographic feature/s. The work is certainly timely, and address an issue that is challenging and 'perhaps' overlooked in the research community.

The paper is well written and structured, the literature review is thorough, and up to date with an interesting critical discussion of various relevant recent studies with some contradicting research reports and findings in the area of covid19 data-related work. The methods used are sound and well presented including good use of illustrations and presentation of tabulated results.

So overall, I think this is a good and timely study, and certainly can be accepted for publications as I believe the research community would benefit from it. That said, I have the following suggestions/ questions:

1. I would have liked to see more discussion of the dataset used? It is stated that the authors used 22 attributes? First, why these 22 only, and second what is the total number of instances in the dataset (did you consider all countries)? I suggest you made this clear at the beginning of the materials section where you discuss the dataset
2. Figure 3 is difficult to read, I wonder if there is a better way to present the results
3. The authors utilised class-association rules and self-organising maps to uncover these 'potential' associations. Is there a reason for this? Why not other machine learning techniques? I suggest just give some justification for these choices

Reviewer: 3

Comments to the Author(s)

The article presents an analysis of the effect of COVID-19 by considering various demographic characteristics. The paper is well-written, but it would probably help readability if a list of the main abbreviations was included in the introduction.

There is an excessive number of percentages in the second paragraph of the introduction. These could be replaced by qualitative observations that explain the timeliness of the present study.

The methodologies presented are sound and adequately explained. However, a wide range of other methodologies could have been alternatively used. For this reason, it would make sense to include in the text some additional explanations regarding the reasons that made the authors choose these specific methodologies.

The authors use publicly available data, collected up to a specific date. It may be worthwhile to consider including additional (recent) data in order to test the consistency of the followed methodological approach and ensure the relevance of the conclusions over time.

The text could become clearer if some observations were expressed in more qualitative terms. For example, instead of stating that "A has negative/positive association with B", one could just state that "when A is higher/lower, B seems to be lower/higher".

Some additional efforts could be made to explain contradictions between the conclusions reached in this present work and conclusions reached in prior works.

===PREPARING YOUR MANUSCRIPT===

===PREPARING YOUR REVISION IN SCHOLARONE===

- An individual file of each figure (EPS or print-quality PDF preferred [either format should be produced directly from original creation package], or original software format).
 - An editable file of each table (.doc, .docx, .xls, .xlsx, or .csv).
 - An editable file of all figure and table captions.
- Note: you may upload the figure, table, and caption files in a single Zip folder.
- Any electronic supplementary material (ESM).
 - If you are requesting a discretionary waiver for the article processing charge, the waiver form must be included at this step.
 - If you are providing image files for potential cover images, please upload these at this step, and inform the editorial office you have done so. You must hold the copyright to any image provided.
 - A copy of your point-by-point response to referees and Editors. This will expedite the preparation of your proof.

- Ensure that your data access statement meets the requirements at <https://royalsociety.org/journals/authors/author-guidelines/#data>. You should ensure that you cite the dataset in your reference list. If you have deposited data etc in the Dryad repository, please only include the 'For publication' link at this stage. You should remove the 'For review' link.
- If you are requesting an article processing charge waiver, you must select the relevant waiver option (if requesting a discretionary waiver, the form should have been uploaded at Step 3 'File upload' above).
- If you have uploaded ESM files, please ensure you follow the guidance at <https://royalsociety.org/journals/authors/author-guidelines/#supplementary-material> to include a suitable title and informative caption. An example of appropriate titling and captioning may be found at https://figshare.com/articles/Table_S2_from_Is_there_a_trade-off_between_peak_performance_and_performance_breadth_across_temperatures_for_aerobic_scops_in_teleost_fishes_/3843624.

Author's Response to Decision Letter for (RSOS-201823.R0)

See Appendix A.

Decision letter (RSOS-201823.R1)

Dear Dr Khan,

It is a pleasure to accept your manuscript entitled "Analysing the Impact of Global Demographic Characteristics over the COVID-19 Spread Using Class Rule Mining and Pattern Matching" in its current form for publication in Royal Society Open Science.

COVID-19 rapid publication process:

We are taking steps to expedite the publication of research relevant to the pandemic. If you wish, you can opt to have your paper published as soon as it is ready, rather than waiting for it to be published the scheduled Wednesday.

This means your paper will not be included in the weekly media round-up which the Society sends to journalists ahead of publication. However, it will still appear in the COVID-19 Publishing Collection which journalists will be directed to each week (<https://royalsocietypublishing.org/topic/special-collections/novel-coronavirus-outbreak>).

If you wish to have your paper considered for immediate publication, or to discuss further, please notify openscience_proofs@royalsociety.org and press@royalsociety.org when you respond to this email.

Appendix A

Associate Editor Comments to Author:

Three reviewers have offered a number of minor recommendations to improve your paper - final acceptance will be contingent on your incorporating these changes into the paper.

Response: Many thanks for providing us this opportunity. We have carefully addressed the comments from all reviewers and provided responses point by point as well as making the corresponding changes within the manuscript.

We believe that the reviewers detailed comments and the way we addressed their critique and suggestions has added much value to the papers' substance and presentation.

Reviewer#1:

- I found the paper well written and technically sound in a timely topic. I enjoyed reading it. I recommend its acceptance.

Response: Many thanks for the positive feedback.

Reviewer #2:

The study raises interesting questions, in particular understanding the covid-19 infection severity and whether it is associated with one or more demographic feature/s. The work is certainly timely, and address an issue that is challenging and 'perhaps' overlooked in the research community.

The paper is well written and structured, the literature review is thorough, and up to date with an interesting critical discussion of various relevant recent studies with some contradicting research reports and findings in the area of covid19 data-related work. The methods used are sound and well presented including good use of illustrations and presentation of tabulated results. So overall, I think this is a good and timely study, and certainly can be accepted for publications as I believe the research community would benefit from it.

Response: Many thanks for the positive feedback.

That said, I have the following suggestions/ questions:

- 1) I would have liked to see more discussion of the dataset used? It is stated that the authors used 22 attributes? First, why these 22 only, and second what is the total number of instances in the dataset (did you consider all countries)? I suggest you made this clear at the beginning of the materials section where you discuss the dataset

Response: Authors are thankful for raising this point. We have updated the text in Section 4.1 (1st paragraph) and 4.2 (1st paragraph).

- 2) Figure 3 is difficult to read, I wonder if there is a better way to present the results

Response: Firstly, this figure is very important and a *unique* way to present the COVID-19 spread across the globe (i.e. in a single 2D figure, one can visualise the COVID-19 severity worldwide).

Following the reviewer's suggestion, we added new figure with better visualisation and quality.

- 3) The authors utilised class-association rules and self-organising maps to uncover these ‘potential’ associations. Is there a reason for this? Why not other machine learning techniques? I suggest just give some justification for these choices.

Response: We understand the point raised by the reviewer however, the proposed work *aims to analyse the patterns and associations instead of prediction or classification where other machine learning techniques may be a choice*. Class rule mining is one of the well-known technique to analyse the frequent patterns within the larger datasets and multiple attributes as in this study. It provides the outcomes as rules which are combinations of frequently appearing attributes that are easily understandable by humans.

Another reason to use rule mining is the analysis of simultaneous combinations of multiple variables which is impractical otherwise. For instance, alternative approach such as multi-variant analysis, could also be used, however, the assumption of data linearity as well as attribute independence would be considered which is not required in rule mining.

We could also use probabilistic models specifically Bayesian however, again, domain knowledge would be needed to form the conditional probability tables of 22 attributes which is not possible in this case. Alternatively, CPT using probabilities could be used however, assumption of attribute independence might not be true.

In addition to rule mining, we used SOM as alternative approach which has TWO benefit in this study. Firstly, we utilise it for validation of rule mining outcomes. Secondly, we used its powerful feature of visualising high dimensional data into 2D (as in Figure 3 and Figure 9) that helps to understand the interdependence of variables as well as patterns distributions.

Finally, following the reviewer’s comment, we highlighted the justification of using these approaches in the related sections (Section 2, Section 4.2). Below are some of the sentences we added:

In Section 2:

“Whilst the aforementioned studies have identified some clinical and economic demographic parameters to predict disease spread and its associations, most of the works are either carried out at early stages with insufficient amount of data, or using conventional statistical approaches, which are limited to investigate the individual attributes’ associations with COVID-19 infection”. An intelligent algorithm is needed to model the complex and multidimensional attributes and investigate the combined impact of various demographic characteristics over the COVID-19 severity, particularly, at the current stage, where sufficient data is available.

In Section 3:

However, the scope of these works is either limited to medical aspects or the analysis of individual association identification, where the outcomes indicated potential contradictions with other works. This might be due to several factors such as immature data/information about COVID-19 in the early stages, use of conventional statistical approaches, and/or limitations in the combined analysis of multiple attributes which is presented in this study. More specifically, ongoing waves and variants of COVID-19 further limits the generalisation of existing similar studies conducted at earlier stages with immature data

In Section 4.2:

“One of the major limitations associated with conventional statistical approaches is the inability to analyse complex patterns within a high-dimensional dataset. This study uses various demographic attributes (as listed in Table 1) with diverse variation and ranges, which are difficult to be analyzed by human experts or conventional statistical approaches, e.g., to draw conclusions from multiple combinations of different attributes. One effective means of dealing with multi-dimensional data visualization is SOM, an unsupervised form of artificial neural networks, performing a non-linear projection of a high-dimensional space onto a lower-dimensional (typically, 2-dimensional) map”

“The two-dimensional map representation is useful for pattern identification within the high dimensional data such as the ones dealt with in this study”.

“In addition to the distance plot of Figure 3, SOM provides heat maps, which is a powerful tool to visualize the individual behaviour of multiple attributes across the map”.

“While SOM produces powerful clustering and rich visual information within the numerical data, further investigation in relation to associations between multiple combinations of demographic characteristics and COVID-19 severity might be helpful to understand the complex patterns and inter-relationships within the categorical dataset. For this purpose, we utilize the special case of conventional rule mining known as Class Association Rules (CARs) [41], where the consequent of a rule contains the target attribute (i.e., death severity in this case). As compared to conventional statistical techniques, CARs has the ability to identify frequently occurring patterns within a larger dataset that can be easily interpreted by humans in the form of rules”.

In Section 6:

“The authors believe that the complex associations and patterns within the multi-dimensional demographic attributes in this work are more comprehensively studied, when compared to the use of classical statistical approaches, reported in most of the existing works”.

Reviewer #3:

- 1) The article presents an analysis of the effect of COVID-19 by considering various demographic characteristics. The paper is well-written, but it would probably help readability if a list of the main abbreviations was included in the introduction.

Response: Many thanks for the positive feedback. We have added the abbreviations within the Introduction section.

- 2) There is an excessive number of percentages in the second paragraph of the introduction. These could be replaced by qualitative observations that explain the timeliness of the present study.

Response: Thanks for the comment. We have made the required changes by re-writing the quantities into qualitative form.

- 3) The methodologies presented are sound and adequately explained. However, a wide range of other methodologies could have been alternatively used. For this reason, it would make sense to include in the text some additional explanations regarding the reasons that made the authors choose these specific methodologies.

Response: We are thankful for the comment and following the suggestion, we have highlighted the justification of using these approaches in the related sections (Section 2, Section 4.2, and Section 6).

- 4) The authors use publicly available data, collected up to a specific date. It may be worthwhile to consider including additional (recent) data in order to test the consistency of the followed methodological approach and ensure the relevance of the conclusions over time.

Response: While authors are aware that updating the dataset will change the entire sequence of patterns (e.g. nodes in SOM plots and visualisations) and hence the textual explanation such as referring to nodes etc., in the Results section.

However, at the same time, authors appreciate the suggestion and worked on it thoroughly with up-to-date dataset as of **8th January 2021**.

This will be very useful for proposed study specifically, making it updated and timely. Interestingly, we did not notice significant changes in the outcomes however, we have updated **all the Figures and Tables** according to the new outcomes and results. It is also interesting to notice that the underlying research question and expected patterns/associations are valid on the updated dataset that indicate the **generalisation** of our approach.

- 5) The text could become clearer if some observations were expressed in more qualitative terms. For example, instead of stating that "A has negative/positive association with B", one could just state that "when A is higher/lower, B seems to be lower/higher".

Response: We are thankful for in depth review of our manuscript. We followed the suggestion and made the required updates in the Discussions, results and conclusion section that might improve the readability of our manuscript.

- 6) Some additional efforts could be made to explain contradictions between the conclusions reached in this present work and conclusions reached in prior works.

Response: We have followed the comment and made the changes in the text in Section 5, Section 6 and where appropriate (as highlighted in the revised manuscript).